# Point-Based Shape Representation Generation with a Correspondence-Preserving Diffusion Model

**Shen Zhu**  iD        SZ9JT@VIRGINIA.EDU

**Yinzhu Jin**  iD        YJ3CZ@VIRGINIA.EDU

**Ifrah Zawar**  iD        EMV5RF@UVAHEALTH.ORG

**P. Thomas Fletcher**  iD        PTF8V@VIRGINIA.EDU

*University of Virginia, Charlottesville, VA, USA*

**Editors:** Accepted for publication at MIDL 2025

## Abstract

We propose a diffusion model designed to generate point-based shape representations with correspondences. Traditional statistical shape models have considered point correspondences extensively, but current deep learning methods do not take them into account, focusing on unordered point clouds instead. Current deep generative models for point clouds do not address generating shapes with point correspondences between generated shapes. This work aims to formulate a diffusion model that is capable of generating realistic point-based shape representations, which preserve point correspondences that are present in the training data. Using shape representation data with correspondences derived from Open Access Series of Imaging Studies 3 (OASIS-3), we demonstrate that our correspondence-preserving model effectively generates point-based hippocampal shape representations that are highly realistic compared to existing methods. We further demonstrate the applications of our generative model by downstream tasks, such as conditional generation of healthy and AD subjects and predicting morphological changes of disease progression by counterfactual generation.

**Keywords:** Point Cloud, Point Distribution Model, Generative Model, Diffusion Model

## 1. Introduction

Recent advances in generative models have demonstrated great capabilities in the biomedical domain (Khader et al., 2023; Gu et al., 2023; Jin et al., 2024; Pinaya et al., 2022; Fontanella et al., 2023). These methods have the ability to generate different modalities of biomedical images with high levels of realism. Generative models hold the promise to address many issues in medical imaging, including data augmentation of small sample imaging studies to improve model training (Khader et al., 2023), aiding in the interpretability of machine learning models (Jin et al., 2024), and counterfactual reasoning (Gu et al., 2023).

However, most of the generative models proposed in the biomedical domain focus on image generation. Less attention has been paid to generative models of anatomical shapes. Analysis of anatomical shapes has long been an important factor in understanding biological processes behind development, aging, and disease progression. For example, Alzheimer's disease (AD) is characterized by atrophy in brain structures, particularly the hippocampus. As shown in Figure 1 on the right, point-based shape representations enable quantification of localized morphological changes between the hippocampi of healthy controls and AD

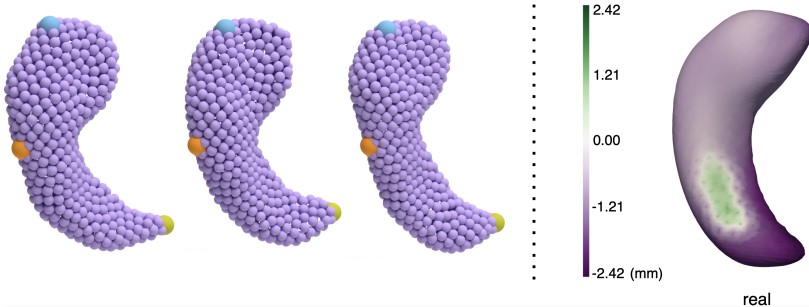

Figure 1: Left: We visualize points at certain indices with blue/orange/green dots respectively. Point-based shape representations of hippocampi maintain correspondences across subjects, with indices reflecting certain anatomical locations; Right: Morphological changes from healthy control to AD. Purple areas show atrophy in most hippocampal substructures.

patients, whereas volumetric information is less precise and will overlook localized morphological changes. Point correspondences between individual shapes are essential for localizing specific morphological differences in anatomy.

Traditional statistical models have long considered correspondences extensively (left of Figure 1), but to the best of our knowledge, no deep generative models have taken correspondences into account. Point cloud generation methods in the computer vision literature (Luo et al., 2021; Zeng et al., 2022) typically consider the point cloud generation process as sampling independently from a point distribution. Luo et al. (2021) use a diffusion model to generate point clouds that are conditioned with a shape latent, which is in turn generated by a normalizing flow. Zeng et al. (2022) propose to use a latent diffusion model with a hierarchical latent space. The sampled latent codes are decoded into a dense point cloud using the trained decoder. Because points within a point cloud are sampled independently, the generated shapes do not have any notion of point correspondences across them. This paradigm is useful when generating common objects like chairs or planes, where correspondence is not important, and we do not need to localize morphological differences.

Motivated by the need for shape correspondences in biomedical applications, we propose a correspondence-preserving diffusion model for point-based shape representations. The backbone of the diffusion model uses a shared linear weights at each layer, modeled after PointNet (Qi et al., 2017). Unlike PointNet, however, we do not desire permutation invariance, as this would destroy the desired point correspondences. Therefore, we add a module in our network to learn correspondence embeddings. Furthermore, to model the spatial relationships among points, we include a masked attention mechanism to share information between neighboring points. We compare our model to the most recent point cloud generative models, and demonstrate the advantages of explicitly modeling correspondences in a shape generation model, using hippocampus data from the OASIS-3 (LaMontagne et al., 2019) dataset.

## 2. Correspondence-Preserving Diffusion Model

We formulate a denoising diffusion probabilistic model (Ho et al., 2020) that converts a noisy point cloud to a meaningful hippocampus representation with correspondences.

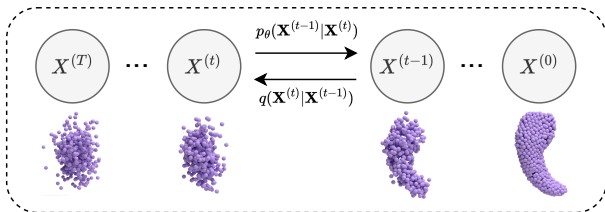

Figure 2: The forward process (lower) and generation process (upper) of the point-based shape representation diffusion model.

### 2.1. Problem Formulation

Previous deep learning based methods for generating point clouds typically do not consider the ordering of points or correspondences. As demonstrated in the shape analysis literature (e.g., Cates et al. (2017); Zhu et al. (2024)), however, correspondences in point-based shape representations can reveal important anatomical information and enable morphological analysis. Motivated by this, we consider a point representation of a shape to be an ordered list of $N$ points, $\mathbf{X}^{(0)} = (x_1^{(0)}, \ldots, x_N^{(0)}), x_i^{(0)} \in \mathbb{R}^3$. A denoising diffusion probabilistic model (Ho et al., 2020) samples from a distribution of data, $p(\mathbf{X}^{(0)})$, as the limit of a reverse diffusion process.

As shown in Figure 2, the diffusion model for point-based shape representations can be divided into two processes. In the *forward process*, Gaussian noise is gradually added to the point positions, and the original shape diffuses into a noisy cloud of points, with a Markov diffusion kernel, defined as

$$q(\mathbf{X}^{(t)}|\mathbf{X}^{(t-1)}) = \mathcal{N}(\mathbf{X}^{(t)}|\sqrt{1-\beta_t}\mathbf{X}^{(t-1)}, \beta_t\mathbf{I}). \tag{1}$$

Model hyperparameters $\beta_1, ..., \beta_T$, referred to as the variance schedule (Ho et al., 2020), control the rates at which noise is added.

The generation process is the reverse of the forward process. It learns denoising transitions $p_\theta(\mathbf{X}^{(t-1)}|\mathbf{X}^{(t)}) = \mathcal{N}(\mathbf{X}^{(t-1)}|\boldsymbol{\mu}_\theta(\mathbf{X}^{(t)}, t), \beta_t\mathbf{I})$ through a neural network. During training, we use a neural network $\epsilon_\theta$ parameterized by $\theta$ to predict the noise $\epsilon$ that is added to the input $\mathbf{X}^{(0)}$. The training objective is to minimize the $L^2$ loss:

$$\mathcal{L}(\theta) = \mathbb{E}_{t,\mathbf{X}^{(0)},\epsilon}\|\epsilon - \epsilon_\theta(\mathbf{X}^{(t)}, t)\|^2, \tag{2}$$

where $t$ is the diffusion timestep.

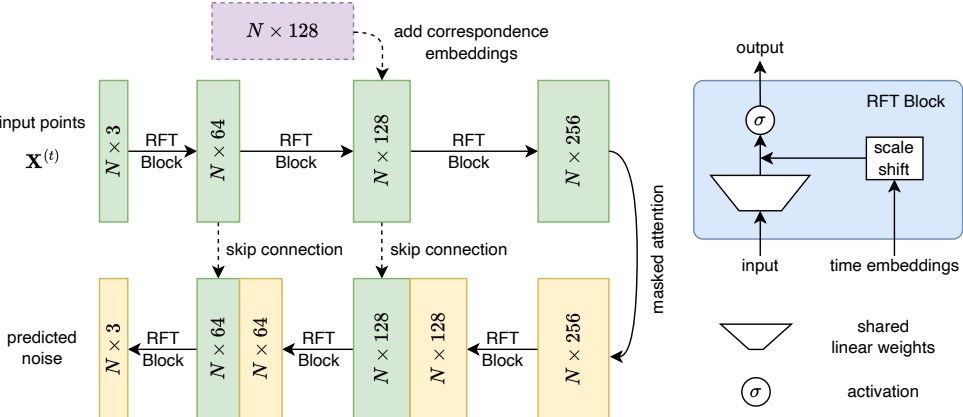

Figure 3: Left: A noisy point set $\mathbf{X}^{(t)}$ is passed through RFT blocks, and then injected correspondence embeddings. Masked attention is used in the bottleneck; Right: the RFT block contains shared linear weights, scale shift, and activation.

## 2.2. Model Architecture

In this section, we delve into the implementation details of our model. Typically, the backbone of diffusion models is implemented using attention U-Net (Oktay et al., 2018). However, the convolutional blocks that are used in such architectures for images are not suitable for point-based shape representations.

Inspired by PointNet (Qi et al., 2017), we propose a U-Net-like architecture with shared linear weights instead of convolutional layers. The architecture is visualized in Figure 3. The input is a noisy point cloud at randomly sampled timestep $t$. At each level, the input is passed through a row-wise feature transformation (RFT) block. We expand the dimension of the points in the encoder of the model with the correspondence embeddings injected.

Previous literature on point clouds does not consider or purposely ignores the ordering of points and correspondences. Inspired by positional embeddings in transformers (Vaswani et al., 2017), we introduce *correspondence embeddings* to encode the index of points. We further demonstrate the effectiveness of it in our ablation study in Section 3.3. Traditionally, sinusoidal positional embeddings are used in transformers to mark the position of tokens in sentences or images. However, these modalities have a natural ordering in them, whereas the ordering in point-based shape representations does not reflect their spatial correlation. Two points adjacent in index might be far apart spatially. Thus, we propose using learned correspondence embeddings, which are more suitable for our scenario. Instead of being fixed like sinusoidal embeddings, correspondence embeddings are learned during training. They are a set of parameters added to the activations in a middle layer, and are updated during training. They act as conditions that we add to the intermediate activations for each point. Suppose the intermediate activations $A = \{y_1, ..., y_N\} \in \mathbb{R}^{N \times z}$, where $N$ is the number of points and $z$ is the dimension for each point $y$. The correspondence embedding $E = \{e_1, ..., e_N\}$ is also of shape $N \times z$. And for point $y_i \in \mathbb{R}^z (i = 1...N)$, the result $y_i'$ after adding the correspondence embedding is obtained by $y_i' = y_i + e_i$.

| | MMD ↓ | | | Coverage↑ | | | Density | | |
|---|---|---|---|---|---|---|---|---|---|
| | CD | EMD | $L^2$ | CD | EMD | $L^2$ | CD | EMD | $L^2$ |
| Luo et al. (2021) | 4.06 | 1.78 | 15.07 | 0.0027 | 0.0064 | $5.4 \times 10^{-4}$ | $4.6 \times 10^{-4}$ | 0.0038 | $7.72 \times 10^{-5}$ |
| Zeng et al. (2022) | **2.42** | 1.34 | 14.26 | 0.17 | 0.078 | 0.0016 | 0.097 | 0.048 | 0.0045 |
| PCA | 2.94 | 1.36 | 1.37 | 0.16 | 0.12 | 0.12 | 1.54 | 1.75 | 1.73 |
| Ours | 2.52 | **1.19** | **1.19** | **0.36** | **0.22** | **0.22** | 0.37 | 0.19 | 0.19 |
| Ablation | 35.11 | 6.18 | 7.95 | 0.003 | 0.003 | 0.003 | 0.04 | 0.05 | 0.03 |
| Real data | 1.42 | 0.81 | 0.82 | 0.98 | 0.99 | 0.99 | 0.96 | 0.97 | 0.97 |

Table 1: Quantitative comparison for point cloud generation. MMD: lower the better, coverage: higher the better, density: the closer to 1 the better. CD, EMD and $L2$ distances are used as the distance measure.

At the bottleneck of the model, the output is passed through a masked self attention layer. The mask for the self attention is generated by selecting the nearest 50 neighbors for each point in the mean shape representation. The masked self attention layer shares the information from the neighboring points with each point, in order to model the spatial relationship. In the decoder of the network, the number of features is gradually reduced at each layer. Skip connections are added between corresponding dimension expanding and reducing RFT blocks.

## 3. Experiments

In this section, we present the dataset and the baseline models used to evaluate our method. Additionally, we outline the metrics adopted for point cloud generation and discuss the corresponding results.

### 3.1. Dataset

The dataset for our experiments comes from OASIS-3 (LaMontagne et al., 2019), which is a compilation of imaging data that consists of participants with dementia and healthy controls. We include 208 AD dementia and 717 healthy control participants. We used ShapeWorks (Cates et al., 2017) to build 512-point shape models for each left and right hippocampus, similar to the process in Zhu et al. (2024).

We perform a 5-fold stratified cross-validation on our dataset based on participants. Each training and test set contains 740 and 185 subjects respectively. We use both the left and flipped right hippocampus as inputs to our diffusion models. Each input shape consists of 512 points.

### 3.2. Baselines

We choose to use principal component analysis (PCA), and two diffusion-based methods by Luo et al. (2021) and Zeng et al. (2022) as baseline methods for comparison to our approach.

For PCA, each shape in the training set is flattened, and the top 128 principal components are selected to capture key variations with minimal information loss. During generation, a random 128-dimensional Gaussian noise vector is sampled and transformed back into data space by reversing PCA. On average, these components explain 99.8% of the variance across the 5-fold cross-validation datasets.

Point cloud generation methods in the computer vision literature (Luo et al., 2021; Zeng et al., 2022) typically consider the point cloud generation process as sampling independently from a point distribution. To the best of our knowledge, no method has taken point correspondences into account. To ensure a fair and meaningful comparison, we use two diffusion-based baselines by Luo et al. (2021) and Zeng et al. (2022) to generate hippocampus point clouds. These methods have proven to be state-of-the-art in this domain. While other methods exist, such as GAN-based and normalizing-flow-based ones (Achlioptas et al., 2018; Yang et al., 2019). Luo et al. (2021) and Zeng et al. (2022) achieve better performance in the experiments in terms of fidelity and diversity. The generation process of Luo et al. (2021) can be considered as sampling from a point distribution, and their method does not consider correspondences between points. The architecture of Zeng et al. (2022) is a latent diffusion model with a hierarchical latent space. The sampled latent codes are then decoded into a dense point cloud using the trained decoder.

### 3.3. Ablation: Importance of Correspondence Embeddings

Using correspondence embeddings might seem redundant when the training data already have correspondence through the ordering of the points. One would think that minimizing the $L^2$ loss alone should keep the generated point-based shape representations in correspondence with each other because $L^2$ distance compares points at the same index. However, our following ablation study showed that in practice, relying on point ordering and $L^2$ loss is not as effective as our proposed approach.

To evaluate the impact of the correspondence embeddings in our proposed model, we conduct an ablation study by removing this component. Table 1 summarizes the performance of the ablation model. It shows that the ablation model struggles to generate points at correct anatomical locations without the assistance of the correspondence embeddings.

### 3.4. Quantitative results

In this section, we will first introduce the metrics that are commonly used for evaluating generated point clouds, followed by the results for each method. Many metrics for evaluating the quality of point cloud generative models have been proposed and used in previous literature (Luo et al., 2021; Zeng et al., 2022). We use the minimum matching distance (MMD), the coverage score, and the density score. Consider a set of real point clouds and a set of generated point clouds. The MMD is calculated as the average of the distance between each real point cloud and its nearest generated point cloud. It measures the *fidelity* of the generated point clouds. Smaller MMD means higher fidelity. Alternatively, one study has proposed using coverage and density to better handle outliers in the data distribution (Naeem et al., 2020). We use the density and coverage in Naeem et al. (2020) with $k = 7$. These two metrics fit a $k$-nearest-neighbor ball to each point in the real dataset. Coverage is defined as the proportion of balls that contain the generated samples. Density is defined

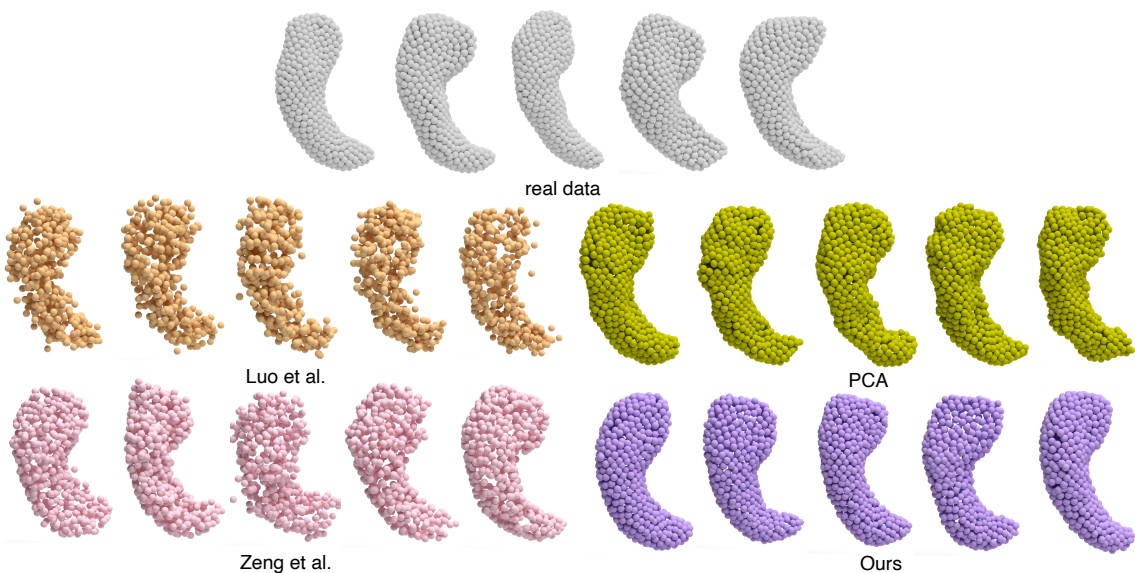

Figure 4: Visualization of the real and generated hippocampal point clouds.

as the total number of generated samples within all balls divided by $kM$ where $M$ is the number of generated samples. High density means more generated samples are located in densely populated regions. The expected values of both metrics between two identical distributions approach 1 as the sample size grows.

Each of these metrics relies on an underlying distance metric between two point clouds. In prior studies, point clouds are unordered, and the distances are defined using permutation-invariant metrics like Chamfer Distance (CD) or Earth Mover's Distance (EMD) (Achlioptas et al., 2018). We include these two distances to ensure fair and reasonable comparison. In order to take correspondences into consideration, we also calculate the metrics using $L^2$ distance in addition to CD and EMD. We recognize that Luo et al. (2021) and Zeng et al. (2022) are not designed to generate point clouds with correspondences, and their performance under $L^2$ distance is expected to be poor. Still, we explicitly include the $L^2$ distance in order to highlight their limitations. The metrics are summarized in Table 1.

We also include the metrics computed between two disjoint real datasets at the bottom of the table for reference. For MMD, our model outperforms other baselines under EMD and $L^2$ distances. Luo et al. (2021) and Zeng et al. (2022) have very large MMD under $L^2$ since the generated samples do not maintain correspondences. Our model outperforms other baselines under all distances for coverage. As for density, PCA has a higher than 1 score, which indicates low diversity in generated samples as reflected in the low coverage score. Our model strikes a good balance between coverage and density compared to other methods.

### 3.5. Qualitative results

We visualize the samples generated by different methods to qualitatively assess the performance of these methods. Additionally, we also showcase that our method is able to

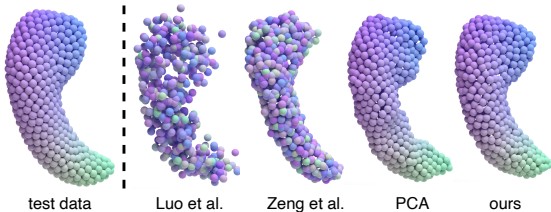

Figure 5: Correspondence between test data and samples from different methods.

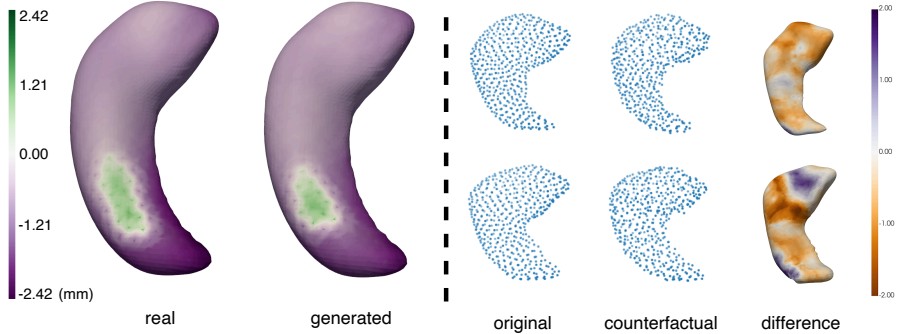

Figure 6: Left: Group difference from healthy to AD. Comparison between real and conditional samples by our method; Right: Original shape representations and their counterfactual. We visualize the change from healthy to AD (orange is atrophy).

generate point-based shape representations with correspondences. Furthermore, we visualize the healthy-to-AD morphological changes to demonstrate that our conditional generative model captures morphological characteristics between classes.

**Generated samples.** We visualize the generated hippocampal point clouds in Figure 4. Samples generated by Luo et al. (2021) and Zeng et al. (2022) typically contain more noise. PCA tends to have less smooth surfaces, and the samples lack diversity. Compared to other methods, hippocampal point clouds generated by our method exhibit more diversity, and are smoother than other methods.

**Correspondence preserving.** To demonstrate the correspondence preserving property of our method, we first obtain a color map based on the mean test data. The color of each point in the mean test data point cloud is determined by its spatial location, as shown in the leftmost figure in Figure 5. We then reuse this colormap to plot samples from different methods. PCA and our methods maintain the correspondence with the test data as the color pattern did not change. However, Luo et al. (2021) and Zeng et al. (2022) failed to capture the correspondences.

**Downstream task.** Point clouds with correspondences can be used to visualize localized morphological changes that cannot be captured by just using volume information, as demonstrated in Zhu et al. (2024), and it's important to maintain the correspondence in order to facilitate this task. To showcase that our model is able to 1) preserve the cor-

respondences and 2) capture the morphological characteristics of healthy and AD groups, we condition our model on class label, and generate the same number of healthy and AD subjects as the training set. We visualize the group mean difference from the healthy group to the AD group in Figure 6 on the left. The patterns look visually similar, showing the potential of our model for modeling fine, localized morphological changes.

Besides, we also perform AD counterfactual generation on healthy samples. On the right of Figure 6, we visualize original healthy samples in the first column, and counterfactual AD samples guided by a trained healthy-AD classifier and similarity to the original samples in the second column. The changes from healthy to AD counterfactual sample are shown in the third column. Orange indicates atrophy from healthy to AD, which is in conformity with the disease progression.

## 4. Discussion

In this work, we introduced a novel approach for generating point-based shape representations with correspondences. Through extensive experiments, we demonstrated its effectiveness in generating shape representations with high fidelity and diversity, and we show its applicability across other anatomical structures.

By providing both qualitative and quantitative insights, we hope this study contributes to the advancement of shape analysis in brain studies. Our preliminary experiments show that the conditioned model is able to capture subtle class differences, and class-guided counterfactual generation produces meaningful shape changes. We would like to apply our method and develop these idea in the future. Additionally, our method currently works in a supervised way with correspondences sorted out. It would be interesting to see how the model performs when we inject noise in the training data. We plan to explore injecting noise and developing a more robust version of the model as part of our future work.

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

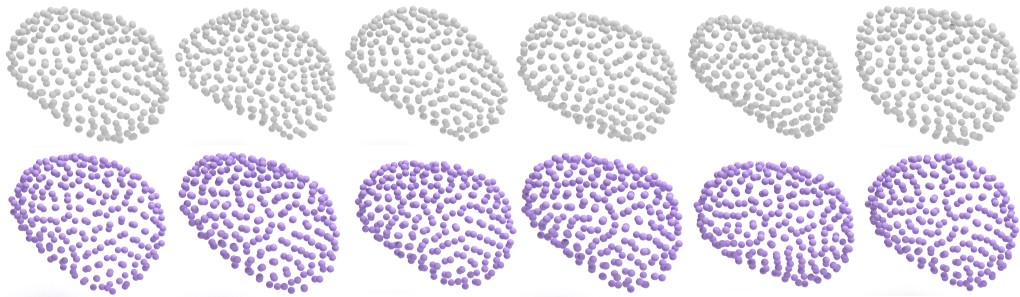

Figure 7: Comparison between real amygdalas (top) and generated amygdalas (bottom).

|  | (Luo et al., 2021) | (Zeng et al., 2022) | PCA | Our method |
|---|---|---|---|---|
| Number of parameters | 3,879,832 | 22,402,731 | 196,608 | 290,883 |
| Training batch size | 64 | 16 | 1480 | 64 |
| GPU peak memory | 4394.44 MB | 15,611.83 MB | - | 9306.46 MB |
| Training epochs | 5,000 | 5,000 | 1 | 5000 |
| Training time | 107.37 mins | $\sim$ 36 h | 10.72 s | 84.80 mins |
| Sampling time | 25.51s | 17.5 mins | 0.02s | 38.74s |

Table 2: Training details and performance measures.

## Appendix A. Amygdala Experiment

To demonstrate the generalizability of our method across different datasets and data collected by different organizations, we conduct experiments on a different anatomical structure and dataset.

We adopt amygdala dataset from National Alzheimer's Coordinating Center (NACC). We also use different numbers of points for the dataset, with 256 points for amygdala. The model architecture is kept the same except for the number of points.

We visualize the original data and generated samples in Figure 7.

## Appendix B. Hippocampus Experiment Details

We list the hippocampus experiment details in Table 2. We train our model and the baselines on a single Nvidia A100 GPU with 80 GB memory. Zeng et al. (2022) has the most parameters and is the most time-consuming to train. Our model has relatively few parameters compared to other deep learning based methods, and the training time is the shortest. Our method strikes a good balance between computational demand and generation quality.

## Appendix C. Model Hyperparameter

We explore the effects of model hyperparameters on the performance, specifically the variance schedule and the number of neighbors to include in the masked attention operation (kNN). We use the same set of metrics as our evaluation criteria.

| | MMD ↓ | | | Coverage↑ | | | Density | | |
|---|---|---|---|---|---|---|---|---|---|
| | CD | EMD | $L^2$ | CD | EMD | $L^2$ | CD | EMD | $L^2$ |
| scaled linear beta schedule | 2.52 | 1.19 | 1.19 | 0.36 | 0.22 | 0.22 | 0.37 | 0.19 | 0.19 |
| sigmoid beta schedule | 3.18 | 1.13 | 1.15 | 0.62 | 0.44 | 0.45 | 0.64 | 0.23 | 0.23 |
| kNN=10 | 3.61 | 1.41 | 1.42 | 0.16 | 0.01 | 0.01 | 0.19 | 0.003 | 0.002 |
| kNN=50 | 2.52 | 1.19 | 1.19 | 0.36 | 0.22 | 0.22 | 0.37 | 0.19 | 0.19 |
| kNN=100 | 3.28 | 1.26 | 1.26 | 0.22 | 0.16 | 0.17 | 0.29 | 0.26 | 0.26 |
| Real data | 1.42 | 0.81 | 0.82 | 0.98 | 0.99 | 0.99 | 0.96 | 0.97 | 0.97 |

Table 3: Quantitative comparison for different hyperparameters.

| | Healthy accuracy | AD accuracy | F1 score |
|---|---|---|---|
| Real test data | 95.8% | 63.4% | 0.71 |
| Conditionally generated data | 97.9% | 53.7% | 0.67 |

Table 4: Downstream task quantitative results.

As shown in Table 3, we experiment with different variance schedules. The sigmoid beta schedule seems to perform better than the linear beta schedule, except for MMD under CD. We also experiment with the cosine schedule, but the results were less than ideal and excluded it from the analysis. We think it's because the default end beta value for the cosine schedule is close to 1, which is too big. Thus, we recommend setting the end beta value to a very small number. We used 0.0205 in our case.

Besides, we also investigated how kNN affects the performance. When setting it to 10, we have very poor coverage and density compared to 50 and 100, meaning the generated shapes are not very realistic or lack diversity. When kNN equals 50, we seem to get the best result. In conclusion, we recommend sigmoid beta schedule, and setting kNN to 50.

## Appendix D. Quantitative Analysis of Downstream Task

In order to measure the performance of the downstream tasks quantitatively, we train a classifier with the training data, and perform classification on both the test data and the conditionally generated data. We keep the size of the test dataset and the conditionally generated dataset to be the same, with the same number of subjects for each label.

The results are shown in Figure 8 and Table 4. We visualize their confusion matrices, and list the accuracies and F1 scores. Based on these results, we think the conditionally generated dataset is able to capture the class differences.

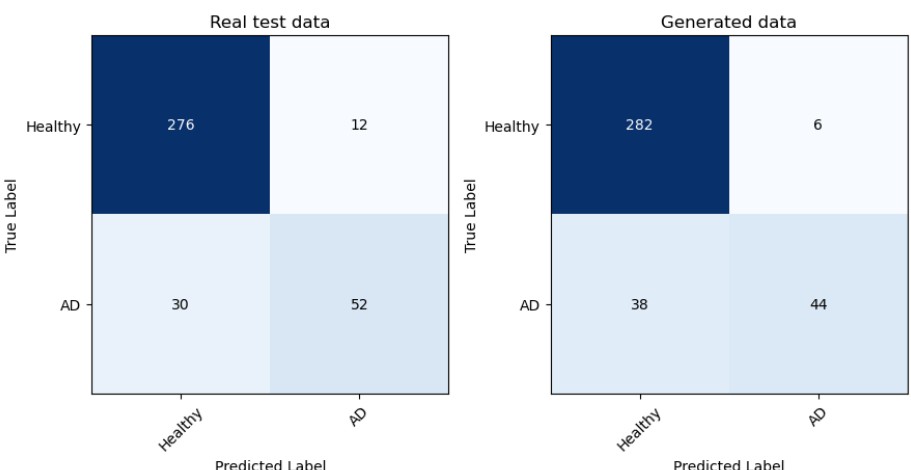

Figure 8: We visualize the confusion matrices for both real test dataset and the generated dataset.

