# OpenReview forum: "Point-Based Shape Representation Generation with a Correspondence-Preserving Diffusion Model"
_MIDL.io/2025/Conference — MIDL 2025 Poster_

### Official Review · Reviewer_yEXp · 2025-02-17

**Confidence:** 4
**Preliminary Rating:** 3
**Recommendation:** Poster
**Final Rating:** 3

**Summary:**

This paper introduces a point cloud diffusion model designed to generate shapes with consistent point correspondences. Specifically, a point with index $i=1, …, N$ in an $N\times 3$ point cloud consistently maps to the same anatomical location across generated shapes. The authors claim that this is particularly interesting for medical applications and state that the problem is not yet addressed in point cloud generation methods from the computer vision literature. Additionally, they show how the proposed model can be used for conditional and counterfactual point cloud generation and claim to outperform existing methods in an unconditional generation task.

**Strengths:**

The main strength of this paper is its focus on an overlooked problem: the generation of correspondence-preserving point clouds. The paper is generally easy to understand and follow. While the proposed approach—adding a learned position embedding to the points—is conceptually simple, it appears to be effective.

The authors compare their method with relatively recent point cloud generation frameworks (CVPR 2021 & NeurIPS 2022) and demonstrate competitive performance. All experiments are performed using publicly available data.

**Weaknesses:**

This paper has several weaknesses:

- The novelty of the proposed method appears to be incremental, as it primarily extends existing point cloud diffusion models by incorporating a learnable correspondence embedding. While this addition may offer some benefits, its contribution relative to prior work is limited.
- The authors should provide more details on how the baseline methods were trained. It is unclear why they perform so poorly. I have quite some experince with both baselines (I used them on several datasets myself) and never generated samples that were somehow as noisy as the samples shown in Figure 4.  Were all methods trained for the same number of epochs? Is the training time comparable?
- The paper lacks information on the computational cost of training and sampling, including GPU memory requirements and overall runtime.
- The paper lacks a discussion section at the end. The authors should provide a concluding discussion that reflects on their findings rather than ending with Section 3.5: Qualitative Results.
- The authors conduct experiments on two downstream tasks (conditional and counterfactual generation) but do not provide any quantitative results. I think this is a weakness.

**Detailed Comments:**

There is a mistake with the timestep indices in the equation that describes the reverse process (Section 2.1). The correct euqation would be $p_{\theta}(\mathbf{X}^{(t-1)}|\mathbf{X}^{(t)})=\mathcal{N}(\mathbf{X}^{(t-1)}|\mu_{\theta}(\mathbf{X}^{(t)},t),\beta_t\mathbf{I})$.

While the introduced loss (Equation 2) is correct, it might be confusing. As you later on talk about $L^2$-Loss (in Section 3.3), I think it would be better to introduce the simplified objective, that is actually used to train these models (MSE).

You should not talk about the "first half" or "second half" of the model as this is very inaccurate. I suggest to reformulate this and use the commonly used expressions for the different network parts: "First half" -> Encoder, "End of first halt" -> Bottleneck, "Second half"-> Decoder.

**Justification Of The Final Rating:**

I appreciate the authors' responses and their efforts to clarify the issues raised by the reviewers. I'm still not fully convinced by the novelty of the work and the experimental results presented in this paper. This is why I will stick with my initial score.

**Justification Of The Preliminary Rating:**

This is a **borderline** paper. While the paper is easy to read and addresses an interesting problem, the novelty is limited. In addition, there are concerns - especially regarding the comparing methods- that need to be addressed before this paper could be published.

**Questions To Address In The Rebuttal:**

The most important point to address in the rebuttal are:

- Why do the comparing methods perform that bad? Can you clarify how the methods were trained in detail.
- Is there a way to quantify the results that are visually presented in Figure 6?

**Special Issue:**

No

---

> ### Author Response · Authors · 2025-03-08
>
> We would like to first thank the reviewer for the thorough and thoughtful feedback. The reviewer's insights have guided us in making meaningful improvements to our manuscript. In the following, we provide a detailed, point-by-point response to each comment, along with the corresponding revisions.
>
> - We appreciate the reviewer’s perspective on the novelty of our work. While our method builds upon existing point cloud diffusion models, the introduction of a learnable correspondence embedding is a non-trivial enhancement. This addition addresses the unique problem that we encounter in the medical image domain, and it improves the model’s ability to capture spatial relationships and enhances performance in scenarios where explicit correspondences are essential.
>          Moreover, our approach differs from prior work in that we don't require training a point cloud encoder, and it's relatively light-weight. We hope this better clarifies the contribution for our approach.
> - We want to thank the reviewer for pointing out adding the training details. To ensure fair and reasonable comparison, we trained our method and the baselines for the same number of epochs. We listed the details in Appendix B. Our model requires the least amount of parameters and training time (excluding PCA), and it's relatively light-weight while maintaining good performance. As for the baseline performance, we hypothesize the small size of medical datasets might have affected the them.
> - We appreciate the reviewer’s suggestion to provide more details on the computational cost of training and sampling. To address this, we have now included detailed information also in Appendix B.
> - Thank you for the suggestion regarding the need for a concluding discussion. In response, we have added a dedicated Discussion section at the end of the paper.
> For the downstream tasks, our primary focus was to demonstrate the qualitative capabilities of our method in conditional and counterfactual generation. However, we acknowledge the value of quantitative evaluation, and we added some quantitative experiments for conditional generation in Appendix D.
> - Thanks for catching our typos and mistakes! We have fixed the one mentioned.
>
>
> The reviewer’s careful analysis has been invaluable, and we are thankful for the perspective it has brought to our work. We would also like to thank the reviewer for correcting some of our notational mistakes. We value the reviewer’s feedback and hope our revisions and clarifications have effectively addressed the raise concerns.

---

> > ### Comment · Reviewer_yEXp · 2025-03-13
> >
> > I would like to thank the authors for their response, but still have an open question:
> > - You write that sampling from LION (Zeng et al., 2022) takes ~1h. This sounds really unrealistic and like you have a problem with your implementation. The original paper reports sampling times of ~27 seconds (in a 1000 step DDPM setup and less than a second in a 25-step DDIM setting). It also remains unclear to me why a latent diffusion model, which performs the time-consuming diffusion process on a compressed latent representation, should have a longer sampling (and training) time than a standard DDPM. So far these added information made my concerns regarding the comparing methods even bigger. Can you clarify this point?
> > - You additionally hypothesize that the small size of medical datasets might have affect the performance of the comparing methods. The LION paper also has some experiments on small size datasets: ShapeNet Mug (149 samples) and ShapeNet Bottle (340 samples) and reports good performance, which is in line with my experience so I can't really believe that the method should perform that bad in a setting with even more training data. Can you also comment on this concern?

---

> > > ### Author Response · Authors · 2025-03-14
> > >
> > > We would like to thank the reviewer for raising these concerns. And these are our responses:
> > >
> > > - We had some inconsistencies in the experiment process and mistakenly used a different batch size configuration for Zeng et al. LION. We sampled using the same batch size as other methods and each batch takes around 105s and in total it’s around 17.5 minutes for LION.
> > > - We looked at Appendix F.5 in the baseline method, and didn’t find the training epoch number. The default training epoch for LION is much bigger than ours. Thus, another possible explanation would be that LION requires more training time. Besides, LION did achieve the best MMD under CD and comparable MMD under EMD in our experiment (Table 1), compared to our method and Luo et al.

---

### Official Review · Reviewer_usmX · 2025-02-20

**Confidence:** 2
**Preliminary Rating:** 3
**Recommendation:** Poster
**Final Rating:** 4

**Summary:**

The paper proposes a novel denoising diffusion probabilistic model tailored for generating point-based shape representations with preserved correspondences—a critical requirement in biomedical shape analysis. Unlike conventional deep generative models that treat point clouds as unordered sets, the proposed method integrates learned correspondence embeddings and a masked attention mechanism into a PointNet-inspired architecture. Experiments on hippocampal shapes derived from the OASIS-3 dataset demonstrate that the model outperforms baseline methods (e.g., PCA, diffusion models by Luo et al. and Zeng et al.) in terms of metrics such as minimum matching distance, coverage, and density, and importantly, it preserves point correspondences necessary for analyzing localized morphological changes.

**Strengths:**

The paper addresses a notable gap in deep generative models by explicitly modeling point correspondences, which is essential for capturing localized anatomical variations. Its strengths include a well-motivated architecture that combines diffusion processes with correspondence embeddings and masked attention, ensuring the generated point clouds maintain a consistent ordering. Extensive quantitative evaluations using metrics like MMD, coverage, and density, along with thorough ablation studies, validate the model's effectiveness. Furthermore, the application to hippocampal shape generation for Alzheimer’s disease analysis underscores its practical relevance in biomedical imaging. Overall, the method represents a significant step forward in generating anatomically meaningful shape representations.

**Weaknesses:**

1. The approach may be computationally intensive compared to simpler methods like PCA, and the scalability to more complex or larger point clouds is not fully addressed.
2. The reliance on high-quality training correspondences raises concerns about performance in scenarios with noisy or imperfect labels. 3. 3. While the experimental validation on hippocampal shapes is promising, the paper would benefit from a broader discussion on the generalizability of the method to other anatomical structures or domains.
4. A more detailed analysis of the sensitivity to hyperparameters—such as the variance schedule—could provide further insights into the model’s robustness.

**Detailed Comments:**

The paper is well-structured and offers a clear exposition of its novel contributions. The integration of learned correspondence embeddings into a diffusion model framework is particularly innovative, as it directly addresses the shortcomings of previous generative models for point clouds. The mathematical formulation is rigorous, and the use of a masked self-attention layer effectively models spatial relationships among points. The extensive ablation studies convincingly demonstrate the necessity of each component, particularly the correspondence embeddings, for maintaining anatomical fidelity. However, more discussion on computational overhead and scalability, as well as comparisons with additional baselines or real-world variability, would further solidify the contribution.

**Justification Of The Final Rating:**

The authors’ rebuttal effectively addresses my key concerns regarding computational overhead, robustness to noisy training correspondences, generalizability, and hyperparameter sensitivity. They provide detailed comparisons of computational cost, additional experiments on the amygdala dataset to demonstrate generalizability, and sensitivity analyses of variance schedules in the appendix, which in my view makes the manuscript much stronger and more comprehensive. Although the model remains computationally more intensive than simpler methods like PCA (as expected), its ability to generate anatomically meaningful, correspondence-preserving point clouds offers a substantial advancement for biomedical shape analysis.

**Justification Of The Preliminary Rating:**

This paper makes a compelling contribution to the field of generative shape modeling by addressing the critical issue of point correspondence preservation in a diffusion model framework. The novel integration of correspondence embeddings and masked attention results in more realistic and anatomically meaningful shape representations, as demonstrated through rigorous quantitative and qualitative evaluations on hippocampal data. However, concerns regarding computational efficiency, scalability, and generalizability limit the overall impact. Addressing these issues—particularly through further analysis of sensitivity to hyperparameters and exploration of other anatomical domains—would enhance the work’s robustness and applicability.

**Questions To Address In The Rebuttal:**

1. Can the authors provide a detailed comparison of the computational cost and runtime of the proposed model relative to simpler methods like PCA or other diffusion models?
2. How robust is the model to potential noise or errors in the training correspondences, and what strategies might mitigate these issues?
3. Is the proposed method readily generalizable to other anatomical structures beyond the hippocampus, or are there domain-specific challenges?
4. Could the authors elaborate on the sensitivity of the model to the choice of hyperparameters, such as the variance schedule and provide guidelines for selecting these parameters?

---

> ### Author Response · Authors · 2025-03-08
>
> Thank you for the reviewer’s thoughtful feedback. The suggestions provided have been invaluable in strengthening our manuscript, and we have worked diligently to incorporate them.
> In response to the reviewer’s suggestions, we have included a detailed comparison of computational overhead in Appendix B, conducted additional experiment on another anatomical structures (amygdala) in Appendix A, and evaluated the model's sensitivity to different variance schedules in Appendix C. Furthermore, we have discussed the potential noise in the training data in the Discussion section.
> - The reviewer requests a detailed comparison of the computational cost and runtime of the proposed model relative to other approaches, especially PCA. We provide the information in Appendix B. PCA is instant compared to other methods, but it has severely limited expressiveness. It only captures linear correlations in the data, making it ineffective for modeling complex, nonlinear structures.
> - The reviewer inquires about the model’s robustness to potential noise or errors in the training correspondences. This is actually a very interesting issue. Currently, our method works in a supervised way with correspondences sorted out. It would be interesting to see how the model performs when we inject noise in the training data. We plan to explore injecting noise and developing a more robust version of the model as part of our future work.
> - As for the generalizability of our work. We have additionally experimented with NACC amygdala dataset. We trained our model in a similar fashion, and visualized the samples in Appendix A.
> - Finally, we added experiments with various noise schedules and number of neighbors in the masked attention. In practice, we noticed that changing model architecture, especially adding the correspondence embeddings, boosted the performance the most. Additionally, using appropriate variance schedule and using moderate number of neighbors for masked attention also help, as shown in the experiments.
>
>
> We sincerely appreciate the reviewer’s thoughtful comments and interesting raised issues, which have been instrumental in improving the quality of our work. We hope that the revisions and clarifications have effectively addressed the raised concerns.

---

### Official Review · Reviewer_u5ZG · 2025-02-21

**Confidence:** 3
**Preliminary Rating:** 4
**Recommendation:** Poster

**Summary:**

The paper focuses on generating realistic shapes through point clouds. In contrast to the prior state of the art, the authors propose to generate an ordered set of points. Such a feature essentially removes the need for shape registration in subsequent analysis.
The authors introduce correspondence embeddings to encode the index of the points. Coupled with the diffusion model, the method archives solid performance. The paper features the ablation experiment and two baseline models.
The main application is the generation of the hippocampus of healthy/Alzheimer's subjects.

**Strengths:**

- the authors focused on a solution to the interesting problem, as generating pre-aligned point sets with known correspondences effectively reduces the workload.
- the method outperforms the baselines in several metrics
- application to downstream task

**Weaknesses:**

This work looks promising, however, it is limited in terms of dataset diversity and the number of baseline methods.
- comparison with the baselines in l2
 two of the baselines were not designed to generate ordered sets of points. is it valid to compare with the baselines as they are expected to perform extremely poorly under l2 metric?

**Detailed Comments:**

I believe this work has the potential to be turned into a good journal paper after substantially improving the validation in terms of dataset diversity and the number of baseline methods.
The paper looks like a good fit for the conference, however, the value of the l2 comparison with the baseline is an open question, as the baselines were not designed in any way to handle the ordered sets, except PCA.

**Justification Of The Preliminary Rating:**

I believe this is a solid application to the medical domain with a unique problem. The paper doesn't carry a rich methodological contribution, however, it focuses on a unique problem, presents a simple solution, and is well-written. I would rate it as weak accept.

**Questions To Address In The Rebuttal:**

validity of comparing the proposed method with the baselines under l2

**Special Issue:**

Yes

---

> ### Author Response · Authors · 2025-03-08
> **New dataset and L2 clarification**
>
> We sincerely appreciate the reviewer’s thoughtful comments and valuable suggestions, which have helped us improve the clarity and quality of our paper.
> Based on the reviewer's suggestions, we have added an experiment using additional dataset in Appendix A and clarified our choice of $L^2$ as one of our metrics in Section 3.4.
>
> - The reviewer finds this work promising but notes concerns regarding the diversity of datasets used and the number of baseline methods included for comparison.
>
>     We agree that demonstrating generalizability across different datasets is important. To address this, we have included an experiment using the **NACC amygdala dataset**, showing that our method is readily applicable to other anatomical structures and data collected from different sites. These results further support the robustness of our approach.
>
>     As for the number of baselines, we carefully selected our baselines to ensure a fair and meaningful comparison, focusing on the most relevant and state-of-the-art methods in this domain.
>   In response to the reviewer’s suggestion, we have clarified our rationale in the revised manuscript in Section 3.2.
>     We welcome any further suggestions on specific baselines the reviewer would find particularly relevant.
>
> - We appreciate the reviewer’s insightful comment regarding the use of the $L^2$ metric for baselines that are not designed to generate ordered sets of points. We recognize that, as expected, the performance of these methods under the $L^2$ metric is likely to be poor. Still, we have explicitly included the $L^2$ metric to highlight their limitations. We have clarified this choice in Section 3.4. Additionally, to have a fair comparison, we also employed the CD and EMD metrics, which are more suitable for scenarios where the generated point sets are unordered.
>
>
> Once again, we are genuinely grateful for the reviewer’s insightful feedback, which has significantly contributed to refining our work. We hope that our revisions and responses have adequately addressed all concerns.

---

### Author Rebuttal · Authors · 2025-03-08

**Rebuttal:**

We greatly appreciate all the effort and valuable feedback that the reviewers have provided. We have make changes in the manuscript accordingly, and the updated version can be found in the Rebuttal Supporting Material. The main changes are as follows:

- Clarification for choice of baselines and $L^2$ metric in section 3.2 and section 3.4
- Discussion section at the end of the paper
- Appendix A: additional dataset experiment
- Appendix B: experiment details and computational overhead
- Appendix C: explore model hyperparameters
- Appendix D: quantitative analysis of one downstream task

Besides, several minor changes have been made to the text according to reviewers' feedback. Mostly typos and notation changes.

**Supporting Material:**

/attachment/66ddb7be720fec88a87a48a465baceca1e5e9d34.zip

---

### Author Response · Authors · 2025-03-12
**Reminder: Author-Reviewer Discussion Period**

**Dear Reviewers**,

We hope this message finds you well! As the author-reviewer discussion period is coming to a close on Friday, we would be grateful for any additional comments you may have on our responses, and we would greatly appreciate any feedback on whether we have adequately addressed your concerns. If there are any remaining questions or points that need clarification, please feel free to reach out before the discussion period ends.

Thank you so much for your time and consideration. We truly value your insights and thoughtful input.

**Best regards,**

**The Authors**

---

### Meta-Review · Area_Chair_aHLr · 2025-03-21

**Recommendation:** Accept (Poster)
**Confidence:** 4

**Metareview:**

The paper presents a correspondence-preserving diffusion model to generate a point-based shape. Two reviewers rated acceptance and one reviewer rated borderline. While the method somewhat lacks novelty, the paper handles the problem of learning shape representation in the medical image domain and analyzes the effect of the diffusion model with explicit correspondences.